# Effect of Dietary Anthocyanin-Extracted Residue on Meat Oxidation and Fatty Acid Profile of Male Dairy Cattle

**DOI:** 10.3390/ani11020322

**Published:** 2021-01-28

**Authors:** Ronnachai Prommachart, Anusorn Cherdthong, Chainarong Navanukraw, Paweena Pongdontri, Wichit Taron, Juntanee Uriyapongson, Suthipong Uriyapongson

**Affiliations:** 1Department of Animal Science, Faculty of Agriculture, Khon Kaen University, Khon Kaen 40002, Thailand; prommachart1@gmail.com (R.P.); anusornc@kku.ac.th (A.C.); chanav@kku.ac.th (C.N.); 2Department of Biochemistry, Faculty of Science, Khon Kaen University, Khon Kaen 40002, Thailand; paweena@kku.ac.th (P.P.); Wichit_t@kkumail.com (W.T.); 3Department of Food Technology, Faculty of Technology, Khon Kaen University, Khon Kaen 40002, Thailand; juntanee@kku.ac.th

**Keywords:** anthocyanin extracted residue, beef, meat oxidation, shelf life, fatty acid profile

## Abstract

**Simple Summary:**

Residue produced by the extraction of anthocyanin from black rice and purple corn contains anthocyanins and phenolic acids. Several researchers have found that anthocyanins and phenolic acids have antioxidant functions in animals. Moreover, black rice and purple extracts have been reported as antioxidants in meat and meat products. However, the effects of anthocyanin-extracted residue (AER) from black rice and purple corn as animal feed on the fatty acid profile and oxidation of meat are still unknown. Therefore, the aim of this study was to investigate the effects of the inclusion level of AER in cattle diet on meat fatty acids’ profile and meat oxidation during storage. Our results showed that AER in cattle diet reduced the oxidation of lipid and protein of meat and had better red color stability during storage. In addition, it could improve concentration of n-3 polyunsaturated fatty acids (PUFA). In summary, AER in the feed of cattle could reduce meat oxidation leading to the extension of the shelf life of meat. Moreover, meat from cattle-fed AER had higher n-3 PUFA, which indicates healthier meat for consumers.

**Abstract:**

This research aimed to evaluate the effects of anthocyanin-extracted residue (AER) in the diet of cattle on meat oxidation during storage and on the fatty acid profiles of the meat. Sixteen male dairy cattle (average body weight 160 ± 10.6 kg) were allotted to feed in a completely randomized design (CRD) with four levels of AER supplementation, 0, 20, 40, and 60 g/kg dry matter (DM) in the total mixed ration (TMR). These TMR diets were fed ad libitum to the cattle throughout the trial. At the end of the feeding trial (125 days), all cattle were slaughtered and meat samples from the *Longissimus dorsi* (LD) muscle were collected to assess meat oxidation and fatty acid profile. The antioxidant effect of AER on meat oxidation was investigated during 14 days of storage based on color, myoglobin redox forms, lipid, and protein oxidation. The results showed meat from cattle fed AER had better color stability, lower oxidation of lipid, protein and myoglobin than did meat from cattle fed the control diet (0 g/kg AER). Furthermore, fatty acid profiles were affected by AER supplementation with an increase in the concentration of n-3 polyunsaturated fatty acids (PUFA). These results support the inclusion of AER supplementation as a natural antioxidant in cattle to reduce meat oxidation and increase PUFA in meat.

## 1. Introduction

Beef consumption in Thailand was about 122,000 tonnes [1] and beef production was about 133,000 tonnes in 2019 [2]. However, there are substantial levels of imports of high-quality frozen beef from Australia and New Zealand [3]. Holstein Friesian (*Bos taurus*) cattle are the most common breed of dairy cattle in Thailand. Dairy farms depend on female cows to produce milk, whereas male calves are a surplus to requirements and farmers are raised for beef production. Advantages of Holstein males for beef fattening are better weights and higher growth performance and meat tenderness compared to *Bos indicus* cattle [4]. Holstein beef would potentially be able to replace imported beef. However, humidity, high levels of solar irradiance, and air temperature of tropical locations negatively affect dairy cattle, causing heat stress in *Bos taurus* cattle [5]. This generates free radicals resulting in oxidative stress [6]. In particular, lipid peroxidation and protein oxidation is generated by oxidative stress, leading to non-microbiological factors related to meat deterioration and low quality [7]. Lipid peroxidation has a negative impact on muscle pigments, which changes in appearance from red to brown influencing the perceived acceptability of the meat among consumers [8]. The process of lipid peroxidation produces off odors, and flavors, cause loss of water-holding capacity and decrease shelf-life [9]. In addition, several reports showed that lipid peroxidation created toxic compounds implicated in a number of human pathologies, including aging processes, atherosclerosis, inflammation and cancer [10]. Moreover, meat from Holstein cattle was sensitive to lipid peroxidation and color deterioration, impacting meat quality [11].

The natural level of resistance to oxidation depends on the balance between free radical production and antioxidant defenses present in meat or muscle tissue, which is the most important internal factor together with fatty acid profiles [12]. The primary strategy to delay or reduce lipid peroxidation in the meat industry is the addition of antioxidants to meat and meat products. The antioxidant substrates are contained in phenolic structures that can be of synthetic or natural origin. However, there are concerns about the potential of synthetic antioxidants to cause toxicological effects [13]. One strategy to reduce meat oxidation relies on the addition of antioxidants from natural supplements to the animal diets. Several researchers reported that feeding with natural antioxidants improve meat oxidation in chickens [14], pigs [15], lamb [16], goats [17] and cattle [18].

Feeding agro-industrial residues to livestock not only reduces environmental problems caused by residue accumulation and decreases the carbon footprints of animal production [19] but also improves the shelf-life stability and quality of meat [16,20] due to the presence of bioactive compounds such as phytochemicals and vitamins. Black rice (*Oryza sativa* L.) and purple corn (*Zea mays* L.) have been described as novel sources of natural antioxidants. They are rich in anthocyanins and phenolic compounds [21,22], which play a role as antioxidant that their structure can be able to donate electrons to the free radicals with unpaired electrons and reduce agents in the electron-transfer reaction pathway [23]. Antioxidant activity of anthocyanin also increased desaturase enzymes activity for converting monounsaturated fatty acid (MUFA) to polyunsaturated fatty acid (PUFA) or inserting additional unsaturated bonds into already existing PUFA [24]. Several studies reported that black rice extract retarded lipid peroxidation, improved redness of color and increased oxidative stability during storage of ground beef patties [25], steak beef [26], pork patties [27], and Thai fermented sausage [28]. Likewise, studies indicated that dietary intake of plant-rich anthocyanin improved oxidative stress status in animals [29,30] and increased the omega-3 (n-3) and omega-6 (n-6) PUFA proportion in the plasma of humans [31], fish [32], and rats [33]. 

Although the addition of anthocyanin extracted from black rice has antioxidant functions in meat and meat products, no studies have been undertaken to investigate the effects of dietary intake of anthocyanin on meat oxidation and the fatty acid profiles of cattle. Therefore, the objectives of this study were to evaluate the effect of dietary supplements of anthocyanin-extracted residue (AER) on the fatty acid profiles and meat oxidation in male dairy cattle.

## 2. Materials and Methods

### 2.1. Animals, Treatments, and Experimental Design

This experiment was conducted at the Department of Animal Science, Khon Kaen University, Thailand (16°28′00.3″ N 102°48′37.9″ E). Animal care has followed the guidelines recommended by the Animal Ethics Committee of Khon Kaen University (U1-04090-2559). Sixteen male Holstein cattle (7–8 months) with an average body weight of 160 ± 10.6 kg, similar body condition score, obtained from the Muak Lek Dairy Co-operative Ltd., Saraburi, Thailand, were randomly assigned to one of four treatment groups (n = 4) according to a completely randomized design (CRD) with varying levels of added AER in the total mixed ration (TMR). The experimental diet groups were labeled T1 (AER 0 g/kg in TMR), T2 (AER 20 g/kg in TMR), T3 (AER 40 g/kg in TMR) and T4 (AER 60 g/kg in TMR), respectively. All TMR diets were formulated to provide 12% crude protein (CP) and 15 MJ/kg gross energy (GE) to meet the nutrient requirements of growing male dairy cattle according to the National Research Council (NRC) [34]. The ingredients, chemical composition and fatty acid profiles of the experimental diets are shown in Table 1. Dry matter (DM), CP, fat content (measured by ether extract, EE) and ash were determined according the method prescribed by the Association of Official Analytical Chemists (AOAC) [35], whereas neutral detergent fiber (NDF) and acid detergent fiber (ADF) were analyzed using procedures described by Van Soest et al. [36]. The GE levels of feed samples were measured using isoperibol bomb calorimeter AC500 (Leco, St. Joseph, MI, USA).

The bioactive compounds of feed and AER were determined by using samples (0.2 g) extracted twice with 10 mL of acidified methanol dissolved in 1.0 N HCl (85:15 *v*/*v*) and incubated at room temperature for 2 h with shaking. The extracts were centrifuged at 3000 rpm at 4 °C for 10 min and the clear supernatants were collected into a volumetric flask.

Total anthocyanin content was determined by the pH-differential method according to Lee et al. [37]. Briefly, two aliquots (0.5 mL) of the extracts were diluted (4.5 mL) with buffer reagents (0.025 M potassium chloride at pH 1.0 and 0.4 M sodium acetate at pH 4.5) and incubated in dark for 20 min at room temperature. The absorbance was measured at 520 and 700 nm respectively, using a UV1280-Vis Spectrophotometer (Shimadzu, Kyoto, Japan). The results were expressed in mg of cyanidin-3-O-glucoside equivalents per g dry weight using the equation:Anthocyanins content (cyaniding-3-glucoside equivalents, mg/L) =A × MW × DF × 10^3^/(ε × 1)(1)
where A = pH1.0 (A_520_ nm − A_700_ nm) − pH4.5 (A_520_ nm − A_700_ nm); MW = molecular weight of cyanidin-3-glucoside (449.2 g/mol); DF = dilution factor; 10^3^ = factor for conversion from g to mg; **ε** = molar extinction coefficient of cyanidin-3-glucoside (26,900 M^−1^·cm^−1^); 1 = the path length (cm).

The total phenolic content in the feed and AER were determined using the Folin–Ciocalteu method [38]. Briefly, 200 μL of extract was mixed with 600 μL Folin–Ciocalteu reagent (10%) and stood at room temperature for 5 min; 500 μL of Na_2_CO_3_ (700 mM) solution was added to the mixture and incubated at room temperature for 2 h. The absorbance was measured at 765 nm via the UV1280-Vis Spectrophotometer (Shimadzu, Kyoto, Japan). Gallic acid was used as the calibration standard, and the total phenolic content was expressed as milligrams of gallic acid equivalent (GAE) per gram of dry weight.

Fatty acid analyses were performed by one-step extraction and transmethylation as described by Ruiz-López et al. [39]. Qualitative measurements of fatty acid methyl esters (FAME) content were performed by gas chromatography using Agilent Technologies 7890B Gas Chromatograph (GC) System (Agilent Technologies, Santa Clara, CA, USA) with fused silica capillary columns (30 m × 0.25 mm, film thickness 0.25 µm; Stabilwax-MS, Bellefonte, PA, USA).

The cattle were kept indoors housing in individual pens (2.5 m × 5 m) with concrete floors and were cleaned every morning. Each pen was equipped with a water trough and a feed bunk. Mean temperature, relative humidity, and rainfall are 24 °C, 64%, and 36 mm, respectively. All cattle were given their assigned TMR diet ad libitum at 7.00 a.m. and 4.00 p.m. each day. The water was available at all times. The cattle were raised for 30 days with TMR diet (AER 0 g/kg in TMR) for adaptation and 125 days for the feeding trial with TMR treatments. At the end of the feeding trail (≈350 kg final weight), all cattle were transported to a local commercial abattoir and fasted for 12 h before slaughter. The cattle were slaughtered according to the conventional procedure that included stunning via gunshot and exsanguination. The carcasses were cut according to the Thai cutting style. Meat samples (100 g) were collected from the portion of *Longissimus dorsi* (LD) muscle, vacuum packed in aluminum foil bags and frozen at −20 °C until the fatty acid profiles could be analyzed. Another section of LD meat samples was taken between the 6th to 12th ribs and kept in the chilling room (2–4 °C) for 48 h before determining meat oxidation. The percentage of intramuscular fat (IMF) of LD determined by Soxhlet instrument using petroleum ether extract [35].

### 2.2. Preparation of Meat Sample and Display Conditions

After chilling, the LD muscle from each carcass was trimmed of external fat and manually cut into piece 1.5 cm thickness (12 pieces for 3 replication). Samples were then randomly assigned to one of four storage times (1, 3, 7, and 14 days of storage) and placed on moisture absorbent pads in polystyrene trays that were overwrapped with oxygen-permeable polyvinyl chloride (PVC) film. The total number of samples taken from the 16 carcasses for physicochemical analysis was 192 pieces. Samples were stored at 4 °C in a cooling room under tube light emitting diodes (LED, PEMCO T8 interlight, Shenzhen, China), with a color temperature of 3000 K and an average lighting intensity of 1.034.2 Lux to simulate the conditions of supermarket display.

### 2.3. Color Measurement

The color of the meat surface was measured at three locations were randomly selected to calculate the average values using a colorimeter CR-400 Chroma Meter (Konica Minolta Holdings Inc., Osaka, Japan) and expressed as CIE (Commission International de I′Eclairage) color coordinates. Lightness (L*), redness (a*) and yellowness (b*) were recorded. Chroma (C*) and hue angle (hº) were calculated from a* and b* color coordinates [40] using the following equations:C*= (a*^2^ + b*^2^)^1/2^
h° = arctan (b*/a*)

For L*, higher values indicated lighter meat; for a*, higher values indicated redder coloring; and for b*, higher values indicated more yellow coloring; higher chroma values indicated more intense red coloring (color saturation); and greater hue angle indicated greater intensity of discoloration.

### 2.4. Relative Myoglobin Characteristics

Myoglobin was extracted from meat samples following the method of Dai et al. [41] with minor modifications. Briefly, 5 g of meat were mixed with 25 mL of phosphate buffer (0.04 M, pH 6.8) and homogenized using ACE-11 homogenizer (Nihon seiki, Osaka, Japan) at 9000 rpm for 30 s in an ice-cold cup and left in the ice bath for 1 h. Then the homogenate was centrifuged at 12,000 rpm at 4 °C for 25 min. The clear supernatant was filtered with a Whatman no. 42 filter paper (GE Whatman; Sigma Aldrich, St. Louis, MO, USA), and the filtrate was measured using a UV1280-Vis (ultraviolet–visible) spectrophotometer (Shimadzu, Kyoto, Japan) with absorbance at 503, 525, 582, and 557 nm to calculate myoglobin forms. The proportions form of metmyoglobin (MetMb), oxymyoglobin (OMb), and deoxymyoglobin (DeoMb) were expressed in percentage (%) using the method of Tang et al. [42] in the following equations:[DeoMb] = CDeoMb/CMb = −0.534R_1_ + 0.594R_2_ + 0.552R_3_ − 1.329
[OxyMb] = COxyMb/CMb = 0.722R_1_ − 1.432R_2_ − 1.659R_3_ + 2.599
[MetMb] = CMetMb/CMb = −0.159R_1_ − 0.085R_2_ + 1.262R_3_ − 0.520
where R_1_ = A582/A525, R_2_ = A557/A525, R_3_ = A503/A525.

### 2.5. Lipid Peroxidation

Meat samples were analyzed for lipid peroxidation by measuring levels of 2-thiobarbituric acid reactive substances (TBARS) using the method of Buege and Aust, [43]. Briefly, about 5 g meat was mixed with 25 mL of thiobarbituric acid (TBA) stock solution (0.375% thiobarbituric acid, 15% trichloroacetic acid, and 0.25 N HCl) and homogenized using ACE-11 homogenizer (Nihon seiki, Osaka, Japan) at 10,000 rpm for 1 min in an ice-cold cup. The homogenate was incubated at 100 °C in a water bath for 20 min, cooled, and centrifuged (Labnet Hermle C0326-K, Wehingen, Germany) at 12,000 rpm for 20 min at 4 °C. The clear supernatant was collected and measured for absorbance at 532 nm using UV spectrophotometer (T80+UV/Vis, PG Instruments Ltd., Leicestershire, UK) by running a blank containing TBA stock solution. The TBARS value was expressed as milligrams of malonaldehyde per kilogram of meat sample (mg MDA/kg) by calculated using a molar extinction coefficient (156,000 M^−1^·cm^−1^) [40] using the following equation:TBARS number (mg MDA/kg) = sample A532 × [(1M TBA chromagen)/156,000] × [(1 mol/L/M] × (0.030 L/5 g meat) × (72.07 g MDA/mol MDA) × 1000 mg/g) × 1000 g/kg)

### 2.6. Protein Oxidation

Protein oxidation in the myofibrillar proteins of the meat samples was evaluated based on the protein carbonyl content using 2,4-dinitrophenylhydrazine (DNPH) as described by Levine et al. [44] with some modifications. Briefly, 3 g of meat were added to 30 mL of phosphate buffer pH 6.5 (20 mM, containing 0.6 M NaCl) and homogenized using ACE-11 homogenizer (Nihon seiki, Osaka, Japan) at 10,000 rpm for 30 s in an ice-cold cup, and four aliquots of 200 µL of homogenate were distributed in Eppendorf tubes. All aliquots were mixed with 1 mL of ice-cold 10% trichloroacetic acid (TCA) for 15 min in ice-baths to precipitate the proteins. The tube samples were centrifuged at 6000 rpm for 10 min at 4 °C to discard the clear supernatant. TCA (10%) was added (1 mL) to the pellets and the procedures described above were repeated. After removing the supernatant, 500 µL of 10 mM DNPH (dissolved in 2.0 M HCl) were added to the pellets of two aliquots and 500 µL of HCl (2.0 M) without DNPH were added to the pellet of two aliquots for blank. All samples were mixed and left in the dark for 1 h, vortexed every 10 min to be derivative. Subsequently, 500 µL of TCA 20% was added to all samples, which were then vortexed, placed in ice baths for 15 min and centrifuged at 8000 rpm for 10 min in 4 °C. The resulting supernatant was discarded. To remove excess DNPH, protein pellets were washed three times with 1 mL of ethanol: ethylacetate (1:1, *v/v*), vortexed and centrifuged at 8000 rpm for 10 min in 4 °C. After each wash, the supernatant was discarded. After the final wash, the samples were left under the hood for 20 min to remove the excess solvent, and the pellets were subsequently dissolved in 1.5 mL of guanidine hydrochloride (6.0 M) in 20 mM phosphate buffer (pH 6.5) and placed in the dark for 30 min and vortexed every 10 min. Finally, the sample solution was centrifuged at 12,000 rpm for 10 min at 4 °C to remove insoluble material. The carbonyl concentration in the samples was measured using spectrophotometry with absorbance at 370 nm and 280 nm using a UV1280-Vis Spectrophotometer (Shimadzu, Kyoto, Japan). The concentration of protein carbonyl was expressed as nmol carbonyl per mg protein (nmol/mg protein) and calculated using the following equation [44]:C_hydrazone_/C_protein_ = [A_370_/ε_hydrazone_,_370_ × (A_280_ − A_370_ × 0.43)] × 10^6^ [nmol/mg protein]
where ε_hydrazone,370_ is 22,000 M^−1^ cm^−1^ and 0.43 = ε_hydrazone,280_/ε_hydrazone,370_.

### 2.7. Meat Fatty Acid Profile

Fatty acid analysis was performed using the one-step extraction and transmethylation process described by Ruiz-López et al. [39] with some modifications. Briefly, LD samples were trimmed off connective tissue and external fat before blending. Triplicated 0.1 g of blended muscle samples were extracted and transmethylated with 4 mL of a mixture containing H_2_SO_4_: methanol: toluene: 2,2-dimethoxypropane (1.2: 24: 6: 3 by volume) and 500 µL of heptane for 2 h of incubation at 80 °C. After cooling, 200 µL of 0.9% NaCl was added to the samples and they were vortex-mixed for 3 min, then centrifuged at 3500 rpm for 3 min. The heptane layer in the upper phase of the 200 µL that contained the fatty acid methyl esters (FAME) was transferred to a vial bottle (1.5 mL) and mixed with 400 µL of heptane for dilution and immediate analysis. Qualitative measurements FAME content were performed by gas chromatography using an Agilent Technologies 7890B Gas Chromatograph (GC) System (Agilent Technologies, Santa Clara, CA, USA) with fused silica capillary columns (30 m × 0.25 mm, film thickness 0.25 µm; Stabilwax-MS, Bellefonte, PA, USA). The injector and detector temperatures were set at 240 °C and 250 °C, respectively. The GC temperature program was as follows: the initially temperature was 170 °C, held for 2.5 min; 10 °C/min until 180 °C, held for 4.5 min; 10 °C/min until 210 °C, held for 4 min; 10 °C/min until 230 °C, held for 6 min. The total run time was 23 min. The injector was set at 0.5 µL of the sample with a split 5:1. Helium was used as a carrier gas with a flow rate of 3 mL/min. The Food Industry FAME Mix 35077 (RESTEK, Bellefonte, PA, USA) was used as an external standard to identified FAME. Fatty acids were expressed as a percentage of total fatty acids that be calculated as follows: total saturated fatty acids (SFA) = myristic acid (C14:0) + pentadecylic acid (C15:0) + palmitic acid (C16:0) + margaric acid (C17:0) + stearic acid (C18:0) + arachidic acid (C20:0) + behenic acid (C22:0); total monounsaturated fatty acids (MUFA) = palmitoleic acid (C16:1) + heptadecenoic acid (C17:1) + Oleic acid (C18:1) + gondoic acid (C20:1); total polyunsaturated fatty acids (PUFA) n − 6 = linoleic acid (C18:2) + dihomo-γ-linolenic acid (C20:3) + arachidonic acid (C20:4); total PUFA n − 3 = α-Linolenic acid (C18:3) + eicosapentaenoic acid (C20:5) + docosahexaenoic acid (C22:6); total polyunsaturated fatty acids (PUFA) = n − 6 + n − 3. The indices of saturation index (SI) was estimated according to Ulbricht and Southgate, [45] and enzyme activities of Δ9-desaturase for C16 fatty acids and C18 fatty acids were estimated according to Malau-Aduli et al. [46]:Δ9 − desaturase 16 = 100 × [(C16: 1cis9)/(C16: (1cis9) + C16: 0)]
Δ9 − desaturase 18 = 100 × [(C18: 1cis9)/(C18: (1cis9) + C18: 0)]
SI = (C14: 0 + C16: 0 + C18: 0)/ΣMUFA +PUFA

### 2.8. Statistical Analysis

The experimental design of meat oxidation was a split-plot with a completely randomized design (CRD). Dietary treatment was the whole plot and storage time was the split-plot. Within the subplot, the meat was assigned to 1, 3, 7, or 14 days of storage. The repeated option in PROC MIXED was used to assess covariance-variance structure among the repeated measures for displaying color data. The most appropriate structure was determined using Akaike and Bayesian information criterion output. Type-3 fixed effects of treatments diet, storage time, and their interactions were analyzed using the mixed procedure of statistical analysis system [47]. The random terms included animal replication, animal replication*treatments diet (error A), and unspecified residual error B. Least squares means for protected F tests (*p* < 0.05) were separated using the probability of difference (pdiff) option and were considered significant at *p* < 0.05. For fatty acid profiles was analyzed through analysis of variance (ANOVA) using the generalized linear model (GLM). Orthogonal polynomials were evaluated to determine linear and quadratic responses to supplementation of AER, with the level of significance at *p* < 0.05.

## 3. Results

The average total phenolic acid and anthocyanin content in AER were 8.23 mg of gallic acid/g and 1.05 mg of cyanidin-3-O-glucoside equivalents/g by dry weight, respectively. Anthocyanin and total phenolic acid content increased with increased levels of AER in the diet treatments, while anthocyanins were not detected in the control sample (0 g/kg AER). The main fatty acids included in the treatment diets and AER were C16:0, C18:1, and C18:2. However, C16:0, C18:0, C18:3, and total PUFA decreased with higher levels of AER in the diet.

### 3.1. Color Parameter

The effects of AER supplementation on the color of LD during storage time are presented in Table 2. Dietary treatment and storage time did not have any impact on lightness (L*, *p* = 0.66). All meat samples showed similar L* on the first day as they did at the end of the storage time. Yellowness (b*) was affected by storage time (*p* < 0.0001) and dietary treatment (*p* = 0.02). Values for b* decreased in the sample for all treatments as storage time increased. However, samples from the 60 g/kg AER group were the most stable and showed the lowest b* value from day 1 to day 7 of storage.

There was highly significant interaction between dietary treatment and storage time for a* (*p* = 0.0007) and chroma (*p* = 0.0472). The a* value for meat samples from all treatment conditions decreased significantly (*p* < 0.05) during 14 days of storage. There was a significant effect on meat samples from the 60 g/kg AER supplementation group. This group showed the lowest a* value on day 1, whereas on day 7 and day 14, control samples (0 g/kg AER) had significantly lower a* (*p* < 0.05) than did samples from all other AER treatment groups.

Chroma values for meat samples from cattle fed 0 g/kg AER decreased significantly (*p* < 0.05) during 14 days of storage. Meat samples from cattle supplemented with 60 g/kg AER showed a significant effect on day 1 and day 3, with the lowest chroma values. However, such significant differences were no longer evident on day 7 and day 14. Hue angle was also affected by dietary treatment (*p* = 0.07). Meat from cattle with 60 g/kg AER in their diet had the lowest hue angle on day 1 and day 3, and the control samples had the highest hue angle on days 7 and 14.

### 3.2. Relative Myoglobin Characteristics

The effects of AER in the diet on myoglobin redox formation in meat samples during storage time are presented in Table 3. The deoxymyoglobin content of samples from all treatments steadily decreased (*p* < 0.0001) over time. Oxymyoglobin content was lowest for control samples and decreased significantly (*p* = 0.0022) with increased storage time. However, levels of oxymyoglobin in meat samples from cattle supplemented with 20 to 60 g/kg AER did not significantly during storage time.

There was significant interaction (*p* = 0.0018) between dietary treatment and storage time for metmyoglobin levels. Metmyoglobin in meat from all of AER treatment groups increased significantly (*p* < 0.05) during 14 days of storage time. For day 1 and day 3, metmyoglobin was not significantly different among treatments. However, for day 7 and day 14, metmyoglobin in the control samples was significantly (*p* < 0.05) higher than in samples for the other treatments.

### 3.3. Lipid Peroxidation

The least square means and standard error for TBARS values for the meat samples during storage time are presented in Figure 1. There was significant interaction (*p* = 0.0002) between dietary treatment and storage time for TBARS values. TBARS values for meat samples from all AER treatments increased significantly (*p* < 0.05) during 14 days of storage. The TBARS values for meat samples from cattle fed diets with 20 to 60 g/kg AER were lower than for the control group on day 3 and day 7 of storage. Meat samples from cattle fed diets with 60 g/kg AER had the lowest TBARS value on day 14 of storage compared with other treatments.

### 3.4. Protein Oxidation

The least square means and standard error for protein carbonyl values for meat samples during storage time are presented in Figure 2. There was no significant interaction between dietary treatment and storage time for protein carbonyl. However, protein carbonyl was significantly affected by dietary treatment and storage time. Protein carbonyl values increased with increased storage time for all meat samples, whereas samples from the AER dietary treatment groups showed significantly lower protein carbonyl values on day 7 and day 14.

### 3.5. Fatty Acid Composition

The effects of AER in the diet on the fatty acid composition of the LD (g/100 g of total fatty) are presented in Table 4. The main saturated fatty acids were C16:0 (20.80–24.13%) and C18:0 (16.18–16.88%). Other medium chain fatty acid (C14:0, C15:0, C17:0) accounted for approximately 3% of total fatty acids. The amount of saturated fatty acids (C14:0, C15:0, C16:0, C17:0, C18:0, C20:0) and total saturated fatty acid content were not significantly different among treatments. Increased levels of AER in the diet were associated with a linear increase in C22:0; however, the increase was not great enough to reach a statistical significance (*p* = 0.06). The total monounsaturated fatty acid content accounted for 33–39% of the total detected fatty acid content detected. 

The major monounsaturated fatty acid in all treatments was C18:1, accounting for 29–34%. Feeding AER treatments did not significantly differ in monounsaturated fatty acids (C16:1, C17:1, C20:1); on the other hand, C18:1 (*p* = 0.05) and total monounsaturated fatty acids (*p* = 0.06) were slightly decreased when the level of AER increased but not reach a statistical difference.

For all treatments, the total polyunsaturated fatty acid content accounted for 16–25% of the total detected fatty acid content. C18:2 (8.50–13.82%) and C20:4 (5.21–7.68%) were the most abundant polyunsaturated fatty acids in the LD samples. Polyunsaturated fatty acid content (C18:2, C18:3, C20:3, C20:4, C20:5) was increased linearly and significantly (*p* < 0.05) with increased levels of AER in the diet.

The total amounts of polyunsaturated fatty acids, total polyunsaturated fatty acids n-3, total polyunsaturated fatty acids n-6, and the PUFA/SFA ratio increased significantly (*p* < 0.05) and linearly with increased AER in the diet. However, total unsaturated, UFA/SFA, MUFA/SFA ratio, n-6/n-3 ratio, saturation index (SI), D9-desaturase 16 and D9-desaturase 18 were similar for all diets.

## 4. Discussion

### 4.1. Effect of Anthocyanin-Extracted Residue (AER) in the Diet on Meat Color

The amount of AER in the diet had an influence on meat redness (a*), yellowness (b*), chroma and hue angle of LD samples until 14 days of storage. These results were consistent with those of the study by de Oliveira Monteschio et al. [18], who explained that the addition of clove and rosemary essential oils, other natural antioxidants, to the diet reduced meat oxidation would cause differences in color.

In addition, the current results indicated that LD samples obtained from cattle fed higher amounts AER had lower values for a*, b*, chroma, and hue, and less intense and vivid color than the control samples at day 1 of storage. Similarly, Jaturasitha et al. [48] reported that feeding purple rice to fatten pigs resulted in lower a* values compared with feeding them white rice. In addition, Jerónimo et al. [49] reported that lamb fed 6% vegetable oil in diets had lower a* values due to high PUFA content in the meat, which was susceptible to high myoglobin oxidation.

The current results showed that on day 3 of storage, a* values for meat from cattle fed AER was not significantly different compared with those of the control group; however, at days 7 and 14, a* values were higher, indicating that myoglobin oxidation was lower in meat from cattle fed AER after 3 days in storage. The reduction in a* values from day 1 to day 14 storage was due to the stage of oxymyoglobin to metmyoglobin and the lipid peroxidation interaction in meat discoloration, leading to increased brown coloring in the meat, which is associated with spoiled and old meat [50]. In agreement with Cardoso et al. [51], these results indicate that meat quality was altered by diet because a* values indicated delayed color deterioration during storage. Antioxidants had the effect of stabilizing the muscle membranes, leading to improvement in meat color [52].

Changes in meat color (a* and b* values) over time in this experiment were due to oxidation of red oxymyoglobin into metmyoglobin, which causes brown coloring in meat [53]. However, meat from cattle fed higher percentages of AER in their diets showed less color deterioration and lower metmyoglobin formation, possibly because of the antioxidant effect of reduction in oxidation. Several studies have reported that anthocyanins as a natural source of antioxidants inhibited both lipid and protein oxidation and stabilized the red coloring in the meat during refrigerated storage [53,54].

### 4.2. Effect of AER in the Diet on Meat Oxidation

The sensitivity of meat to oxidation can vary by animal species and breed and muscle type assessed, as well as by the diet provided to the animals. This study found that the lipid peroxidation (TBARS) and protein oxidation (protein carbonyl) at days 3 to 14 of storage in cattle fed diets supplemented with AER were lower than in cattle fed without AER. This difference was associated with high dietary contents of anthocyanins and phenolic compounds, which may improve the oxidative stability of meat over time. However, over feeding antioxidant supplements to animals may cause pro-oxidant effects and increase oxidation [55].

The effect of antioxidant supplementation in the diet was consistent with effects observed in previous studies [48,52], whereby black rice was fed to pigs and it was reported that TBARS content in raw meat reduced over time in storage. Chikwanha et al. [56] reported a reduction in lipid and protein oxidation of raw meat when grape pomace was added to the diet of lambs. de Oliveira Monteschio et al. [18] demonstrated that supplementation of natural substances in the diets of feedlot-finished heifers decreased lipid peroxidation in raw meat during aging. TBARS values in this study ranged from 1.57 to 1.91 mg MDA/kg on day 14 of storage, which may reflect increased risk of off-flavor development in the meat. According to Campo et al. [57], 2 mg MDA/kg of meat could be considered the threshold indicating off-flavor in oxidized beef. An imbalance between antioxidant ability and free radicals in meat may increase the risk of oxidative damage [58]. The metal chelation process and consumption in the chain breaking mechanism probably cause the reduction of antioxidants, resulting in increased levels of oxidation [59]. Several reports confirmed that low oxidation values in the meat corresponded with the high antioxidant activity [25,56,60]. Higher levels of lipid peroxidation and protein carbonyl content may be related to exposure to pro-oxidant factors [61].

Increases in protein carbonyl have been shown to determine protein oxidation in meat. Protein oxidation can have deleterious effects on meat quality such as reduced water-holding capacity [62], altered color [62], texture [63], flavor [64], aroma [65], and nutritional value [62]. The total protein carbonyl content in animal tissue is estimated to be 1–2 nmol/mg protein [65]. However, protein carbonyl increases during aging/chilled storage of meat [62]. The level of protein carbonyls reported in this study was consistent with Rowe et al. [66], who found that chilled storage of beef for 10 days increased protein carbonyl from 3.1 to 5.1 nmol/mg protein, and Lindahl et al. [67], who reported an increase from 4.8 to 6.9 nmol/mg protein. However, this study showed that AER supplementation can slow increases in protein carbonyl after day 7 of storage, which supports the findings of Ganhão et al. [68] who found that phenolic-rich fruit extract as an antioxidant source may protect against the formation of protein carbonyl in cooked patties.

### 4.3. Effect of AER in the Diet on Fatty Acid Profile

The ratio of saturated and unsaturated fatty acids in meat samples in this experiment was similar to that of previous research with crossbred bulls fed essential oils [55], with steers fed wet distillers grains plus solubles [69], with bulls fed concentrate and TMR [70], and with steers fed algal residue [71]. However, there were differences in the fatty acid composition of n-3, n-6, PUFA and PUFA/SFA among treatments.

In previous studies, dietary intake of plant-rich anthocyanins increased the n-3 and n-6 PUFA proportion in the plasma of humans, fish, and mammalian models such as rats. de Lorgeril et al. [31] reported that proportions of n-3 PUFA in plasma were increased in red wine drinkers because red wines are rich in polyphenols. Villasante et al. [32] found that feeding purple corn extract rich in anthocyanins increased the proportions of total n-3 PUFA in the plasma of rainbow trout. Similarly, Toufektsian et al. [33] revealed that feeding purple corn extract rich in anthocyanins increased the proportion of very long-chain (n-3) eicosapentaenoic acid (C20:5, EPA) and docosahexaenoic acid (C22:6, DHA) in the plasma of rats. According to Muíño et al. [72], lamb fed diets supplemented with red wine extract rich in proanthocyanidins increased the proportion of EPA in LD muscle. Furthermore, Jaturasitha et al. [48] reported that feeding purple rice to pigs resulted in a higher proportion of n-3 PUFA in raw loin chops.

This study showed that increased levels of AER in the diet were associated with linear increases in the proportions of n-3 and n-6 PUFA. These results indicate that the bioactive compounds such as anthocyanins and phenolic acid in AER may have direct impacts on the proportions of n-3 and n-6 PUFA in beef. Toufektsian et al. [33] suggested that there are two kinds of EPA and DHA enhancements in the body. Anthocyanins may have an impact on the conversion biosynthesis pathway from their precursor α-linolenic acid into EPA and DHA.

AER contained black rice bran that was high in C18:1. Particularly, the proportions of C18:1 in the diet increased following higher AER inclusion. However, the proportions of C18:1 in the meat linearly decreased with higher levels of AER. A similar result was observed by De Mello et al. [69], who reported that beef from steer fed higher distillers grains had proportions of C18:1 linearly decreased. The reason for this decrease is still unknown. It is possible that the anthocyanin from AER may alter ruminal bacteria’s ability in reducing cis n-9 fatty acid, as well as mechanisms of absorption and transportation of these fatty acids from the small intestine to the muscle [73]. The bacteria are largely responsible for biohydrogenation of unsaturated fatty acid in the rumen. Particularly, some bacteria utilized C18:1 as one of the main substrates that is susceptible to rapid hydrogenation, with 18:0 being the end product [70]. Previous studies have reported that increased intake of C18:1 from rice bran oil [74] and distiller dried grains with solubles [75] of steer did not have an effect on C18:1 in meat.

The composition of fatty acids in meat could affect its nutritional value for humans. Consumption of more PUFA than SFA reduces risk factors associated with cardiovascular disease [76]. Thus, the ratio of PUFA/SFA can be used as a measurement of fatty acid composition in food. According to the Annual Report of the National Food Survey Committee (HMSO) [77], a ratio of PUFA/SFA higher than 0.45 is recommended for reducing the risk of cardiovascular disease. High levels of AER in the diet improved the ratio of PUFA/SFA in the treated samples to 0.61 compared with 0.37 for the control samples.

High proportions of n-3 and n-6 PUFA have been used to qualify the nutritive values of beef. The essential fatty acids n-3 and n-6 PUFA must be obtained from food because they cannot be produced de novo. Higher levels of n-3 PUFA and an n-6/n-3 ratio of 4:1 in food is associated with health benefits and a reduction in risk factors related to cardiovascular disease and cancer [78]. The addition of AER did not increase the n-6/n-3 ratio; however, the ratios for all treatments groups were higher than 4:1 due to a high proportion of linoleic acid (C18:2) from cereals (corn) used in the feed, resulting in high a proportion of linoleic acid in the beef. According to Fruet et al. [79], meat from grain maize-fed animals had higher n-6/n-3 ratios than meat from pasture-fed animals because grains contain high levels of linoleic acid (n-6 fatty acid).

## 5. Conclusions

Feeding diets containing AER affected the quality of meat during storage time. Meat from cattle-fed diets containing AER, showed greater color stability and lower oxidation of lipid, protein and myoglobin than meat from cattle fed diets without AER. These results suggest the conclusion that antioxidants in the feed of cattle improve meat oxidation leading to an extension of the shelf life of meat. Moreover, meat from cattle fed AER had a higher proportion of n-3 PUFA, which may indicate the meat is healthier for consumers; however, more research examining the total amount of n-3 PUFA in meat is also required in order to determine possible beneficial effects.

## Figures and Tables

**Figure 1 animals-11-00322-f001:**
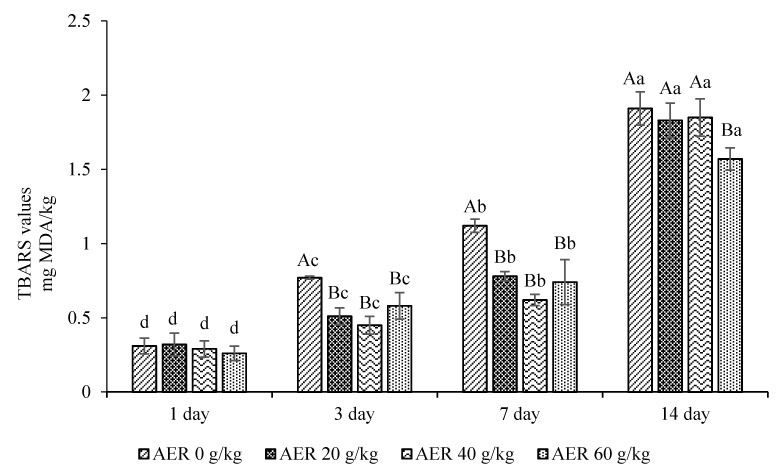
Thiobarbituric acid reactive substances (TBARS) values (mg malonaldehyde (MDA)/kg) of LD from male dairy cattle with AER in TMR diet at different storage time. ^A–B^ Least square means with different letters within the same day of storage time were significantly different (*p* < 0.05); ^a–d^ Least square means with different letters within the same treatments were significantly different (*p* < 0.05); AER: anthocyanin extracted residue; LD: *Longissimus dorsi*; TMR: total mixed ration.

**Figure 2 animals-11-00322-f002:**
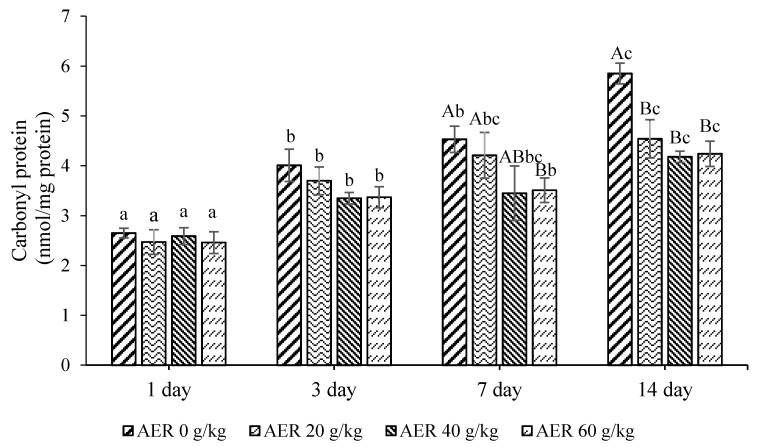
Protein carbonyl of LD from male dairy cattle with AER in TMR diet at different storage time. ^A–B^ Least square means with different letters within the same day of storage time are significantly different (*p* < 0.05); ^a–c^ Least square means with different letters within the same treatments are significantly different (*p* < 0.05); AER: anthocyanin extracted residue; LD: *Longissimus dorsi*; TMR: total mixed ration.

**Table 1 animals-11-00322-t001:** Ingredients, chemical compositions and fatty acid profile of total mixed ration (TMR) diets.

Items	AER, g/kg DM				AER
	0	20	40	60	
**Ingredients (g/kg DM)**					
Cassava pulp	300	300	300	300	
King Napier grass silage	243	243	243	243	
Palm kernel meal	140	140	140	140	
Cassava chip	110	110	110	110	
Dried distillers corn grains with solubles	80	80	80	80	
Defatted rice bran	100	80	60	40	
Anthocyanin extracted residue (AER)	-	20	40	60	
Salt	5	5	5	5	
Sulfur	2	2	2	2	
Dicalcium phosphate	5	5	5	5	
Premix	5	5	5	5	
Urea	10	10	10	10	
Chemical compositions					
Dry matter (g/kg fresh weight)	387	392	395	385	938
Organic matter (g/kg DM)	937	938	939	939	955
Crude protein (g/kg DM)	124	128	123	125	137
Ether extract (g/kg DM)	23	27	33	36	128
Ash (g/kg DM)	63	62	61	61	55
Neutral detergent fiber (g/kg DM)	43.2	43.7	42.4	42.3	32.5
Acid detergent fiber (g/kg DM)	24.6	23.8	24.1	23.9	11.0
Gross energy (MJ/kg DM)	15.23	15.13	15.29	15.20	17.29
Anthocyanin (mg/g DM)	ND	0.02	0.04	0.05	1.05
Total phenolic acid (mg gallic acid/g DM)	2.67	2.80	3.01	3.31	8.23
Fatty acids, g/100 g of total fatty acids					
Palmitic C16:0	23.58	22.95	22.58	22.51	20.42
Palmitoleic C16:1	0.16	0.12	0.17	0.17	0.20
Stearic C18:0	5.10	4.85	4.58	4.25	2.72
Oleic C18:1	33.94	35.07	35.71	36.61	42.01
Linoleic C18:2 n-6	30.71	30.62	30.72	30.34	29.97
α-Linolenic C18:3 n-3	3.37	2.98	2.81	2.58	0.95
Total saturated fatty acids	29.96	29.2	28.57	28.20	24.6
Total monounsaturated fatty acids	35.25	36.39	37.08	38.12	43.70
Total polyunsaturated fatty acids	34.60	34.24	34.18	33.58	31.59
n-6: n-3 fatty acid ratio	9.12	10.28	10.93	11.76	31.7

AER: anthocyanin extracted residue; ND: not detected; DM: dry matter.

**Table 2 animals-11-00322-t002:** Color of *Longissimus dorsi* (LD) from male dairy cattle with AER in TMR diets at different storage time.

Color	AER, g/kg	Storage Time (Day)			
		1	3	7	14
	0	41.15	40.71	40.34	41.32
L *	20	40.25	39.65	39.82	38.92
SE = 2.141	40	41.11	42.22	39.83	38.49
	60	38.19	38.52	37.95	38.76
	0	22.07 ^A,a^	21.05 ^a^	16.04 ^B,b^	14.59 ^B,c^
a *	20	22.83 ^A,a^	21.67 ^a,b^	20.65 ^A,b^	17.62 ^A,c^
SE = 0.899	40	22.35 ^A,a^	21.50 ^a^	19.82 ^A,b^	16.31 ^A,c^
	60	20.14 ^B,a^	19.11 ^a^	18.56 ^A,a^	17.78 ^A,b^
	0	9.68 ^A,a^	9.18 ^A,a^	8.17 ^A,a,b^	6.64 ^b^
b *	20	9.67 ^A,a^	8.97 ^A,a^	8.37 ^A,a^	5.7 ^b^
SE = 0.743	40	8.88 ^A,a^	9.57 ^A,a^	7.55 ^A,a^	5.68 ^b^
	60	6.69 ^B^	6.48 ^B^	6.11 ^B^	6.45
	0	24.12 ^A,a^	22.98 ^A,a^	19.04 ^B,b^	17.24 ^b^
Chroma	20	24.81 ^A,a^	23.49 ^A,a^	22.31 ^A,b^	18.55 ^c^
SE = 1.216	40	24.1 ^A,a^	23.58 ^A,a^	21.27 ^B,b^	17.36 ^c^
	60	21.24 ^B^	20.22 ^B^	19.61 ^B^	18.99
	0	23.71 ^A^	23.48 ^A^	26.81 ^A^	24.48 ^A^
Hue angle	20	22.86 ^A,a^	22.25 ^A,a^	21.9 ^B,a^	17.48 ^B,b^
SE = 2.408	40	21.44 ^A^	23.76 ^A^	20.56 ^B^	19.03 ^B^
	60	18.02 ^B^	18.34 ^B^	17.65 ^B^	19.6 ^B^

^A–C^ Least square means with different letters within the same day of storage time were significantly different (*p* < 0.05); ^a–b^ Least square means with different letters within the same treatments were significantly different (*p* < 0.05); L*: lightness; a*: redness; b*: yellowness; SE: standard errors; AER: anthocyanin extracted residue; LD: *longissimus dorsi*; TMR: total mixed ration.

**Table 3 animals-11-00322-t003:** Proportions of oxymyoglobin (OMb), deoxymyoglobin (DMb), and metmyoglobin (MMb) of LD from male dairy cattle with AER in TMR diets at different storage times.

Parameter	AER, g/kg	Storage Time (Day)			
		1	3	7	14
	0	36.77 ^a^	31.78 ^b^	28.38 ^b,c^	23.85 ^c^
% DMb	20	35.23 ^a^	28.93 ^b^	28.10 ^b^	23.39 ^c^
SE = 2.268	40	35.01 ^a^	33.84 ^a,b^	29.40 ^b^	23.88 ^c^
	60	38.78 ^a^	30.78 ^b^	30.40 ^b^	24.48 ^c^
	0	53.18 ^a^	53.62 ^a^	42.96 ^B,b^	42.12 ^B,b^
% OMb	20	54.56	53.31	51.23 ^A^	50.49 ^A^
SE = 3.699	40	52.91	50.66	47.53 ^A,B^	47.71 ^A,B^
	60	53.36	52.85	48.42 ^A,B^	46.60 ^A,B^
	0	10.93 ^d^	15.59 ^c^	29.59 ^A,b^	34.74 ^A,a^
% MMb	20	11.24 ^d^	18.81 ^c^	21.69 ^B,b^	27.02 ^B,a^
SE = 2.121	40	12.97 ^d^	16.47 ^c^	24.08 ^B,b^	29.21 ^B,a^
	60	9.41 ^d^	17.38 ^c^	22.13 ^B,b^	29.97 ^B,a^

^A–B^ Least square means with different letters within the same day of storage time were significantly different (*p* < 0.05); ^a–d^ Least square means with different letters within the same treatments were significantly different (*p* < 0.05); SE: standard errors; AER: anthocyanin extracted residue; LD: *Longissimus dorsi*; TMR: total mixed ration.

**Table 4 animals-11-00322-t004:** Fatty acid (FA) composition (% of all fatty acids analyzed) of the lipids in LD of male dairy cattle with AER in TMR diets.

Fatty Acids, g/100 g	AER Supplemented, g/kg				SEM	Contrast	
of Total Fatty Acids	0	20	40	60		L	Q
Saturated FA (SFA)							
C14:0	2.88	2.28	3.36	2.47	0.370	0.92	0.71
C15:0	0.05	0.14	0.15	0.11	0.055	0.44	0.24
C16:0	23.50	22.48	24.13	20.80	0.844	0.11	0.19
C17:0	0.52	0.53	0.50	0.48	0.045	0.47	0.72
C18:0	16.18	16.88	16.48	16.81	1.080	0.76	0.86
C20:0	0.18	0.21	0.17	0.27	0.048	0.31	0.45
C22:0	0.19	0.21	0.21	0.30	0.038	0.06	0.37
Other SFA	0.49	0.59	0.49	0.60	0.101	0.58	0.95
Total SFA	43.99	42.79	45.49	41.14	1.595	0.42	0.34
Monounsaturated FA (MUFA)							
C16:1 cis-9	3.13	2.28	2.75	2.03	0.339	0.08	0.87
C17:1 cis-10	0.81	0.82	0.65	0.84	0.076	0.81	0.24
C18:1 cis-9	34.26	30.91	29.89	29.31	1.615	0.05	0.41
C20:1 cis-11	0.20	0.20	0.21	0.21	0.036	0.88	0.94
Other MUFA	1.05	0.83	1.05	0.89	0.222	0.79	0.91
Total MUFA	39.44	35.04	34.56	33.27	2.144	0.06	0.46
Polyunsaturated FA (PUFA)							
C18:2 n-6	8.50	11.85	11.09	13.82	0.861	<0.01	0.72
C18:3 n-3	0.42	0.60	0.56	0.72	0.040	<0.01	0.63
C20:3 n-6	1.32	1.71	1.71	1.89	0.145	0.05	0.74
C20:4 n-6	5.21	6.59	5.88	7.68	0.566	0.01	0.69
C20:5 n-3 (EPA)	0.21	0.34	0.31	0.38	0.035	0.01	0.49
C22:6 n-3 (DHA)	0.30	0.32	0.30	0.39	0.033	0.19	0.29
Other PUFA	0.60	0.66	0.38	0.69	0.172	0.99	0.23
Total PUFA	16.57	22.17	19.95	25.59	1.522	<0.01	0.98
Total PUFA n-3	0.94	1.26	1.18	1.50	0.091	<0.01	0.96
Total PUFA n-6	15.03	20.15	18.39	23.41	1.455	<0.01	0.97
Total UFA	56.01	57.21	54.51	58.86	1.549	0.29	0.2
UFA/SFA	1.27	1.33	1.21	1.43	0.078	0.34	0.33
PUFA/SFA ratio	0.37	0.50	0.43	0.61	0.040	<0.01	0.60
MUFA/SFA ratio	0.90	0.83	0.77	0.81	0.067	0.28	0.41
n-6/n-3 ratio	15.96	16.09	15.64	15.56	0.804	0.71	0.94
Saturation index (SI)	0.45	0.45	0.50	0.43	0.033	0.98	0.28
D9-desaturase 16	11.66	9.39	10.24	9.13	1.171	0.22	0.63
D9-desaturase 18	68.00	64.50	64.35	63.61	2.531	0.26	0.63
IMF content (%)	5.36	4.85	5.83	4.03	0.650	0.32	0.34

EPA: eicosapentaenoic acid; DHA: docosahexaenoic acid; SEM: standard error of means; L: linear; Q: quadratic; AER: anthocyanin extracted residue; LD: *Longissimus dorsi*; TMR: total mixed ration; IMF: intramuscular fat.

## Data Availability

The data presented in this study are available on request from the corresponding author.

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
