# Peer review of "Effect of Dietary Anthocyanin-Extracted Residue on Meat Oxidation and Fatty Acid Profile of Male Dairy Cattle"

_animals, 2021, doi:10.3390/ani11020322_

Round 1
Reviewer 1 Report
I believe that the study is good and well designed. There are parts in the manuscript where English grammar could be more improved. This paper could be suitable for publication while some improvements as mentioned in the comments below have been made.
Specific comments:
Line 46, recommend placing ‘cause’ instead of ‘causes’
Line 46, replace ‘decreases’ with ‘decrease’
Line 62, recommend removing ‘reduce’ and put ‘reduces’
Line 82, …at ‘THE’ Department of Animal Science…
Line 84, …by ‘THE’ animal...
Line 85, correct the word, it’s probably a mistake, …(7-8 months)
Line 87, recommend putting ‘THE’ between words in and total
Line 94, put ‘THE’ between according and method
Line 142, replace ‘piece’ with ‘pieces’
Line 182, instead of ‘were’ put ‘was’
Line 290, put ‘the’ between words in and sample
Line 302, put ‘a’ between words showed and significant
Line 323, replace ‘were’ with ‘was’ (like singular)
Line 324, replace ‘were’ with ‘was’
Line 370, put ‘the’ between words of and total
Line 374, put ‘the’ between words when and level
Line 379, instead of ‘were’ put ‘was’
Line 408, replace ‘were’ with ‘was’ (like singular)
Line 433, replace ‘were’ with ‘was’
Line 449, replace ‘determined’ with ‘determining’
Line 454, at the beginning of the sentence, is recommended to be inserted ‘The level…’
Line 454, replace ‘were’ with ‘was’
Line 488, put ‘the’ between words through and intestinal
Line 512, put ‘the’ between words extend and shelf
Author Response
Thank you for your comments and suggestions. We appreciate your responsibility as a reviewer and assistance with our manuscript. You have provided several good suggestions that are very helpful for revising and improving our manuscript, as well as providing significant guidance to our research. We have studied comments carefully and have revised to the best of our ability. Please see the attachment.

Reviewer 2 Report
Introduction:
Meat from Holstein cattle was sensitive to lipid oxidation and color deterioration,
impacting meat quality [5]. Therefore genetics influence the levels of oxidation, why do the authors choose this breed? include this in materials and methods
the authors state that: “consumers more frequently reject Holstein beef cattle compared with other breed beef” they should write down the bibliographic reference of that statement. In my experience, consumers do not know the origin (breed) of the meat they consume.
There are many other important factors besides the composition of the diet that influence the oxidation of meat, for example stress and feed conditions. It is necessary to talk about them at least in the introduction, despite the fact that in the work only the composition of the feed is valued
Material and methods:
The work is based on a series of laboratory analyzes of samples from 16 animals. The methodology of the laboratory tests is very well described, but the characteristics of the animals and breeding conditions are not just described in the work.
About animals only the following is written:
Sixteen male Holstein cattle (7-8 mouths) with an average body weight of 160±10.6 kg
It is necessary to know:
1.- the genetic characteristics of the animals. Individuals should be chosen as similar as possible (for example the same parent).
2.- the method of slaughter of the animals.
four animals per experimental group seem "a priori" little number of animals for the paper. It would be important for the authors to statistically justify that with four animals per group the results are consistent, or if not they should talk about an initial approximation to the objective of the work.
Author Response

(The authors gave the same response as above.)

Reviewer 3 Report
Congratulations to the authors, who contributed to science. The information presented in this manuscript is excellent, despite the small number of animals used; but as the objective was restricted to meat only; this reviewer understands that the paper can be published. here are just small points to improve:
1) in the introduction section, the authors must give a "technical" justification as to why they believe that anthocyanin would be capable of altering meat, mainly fatty acid profile. Authors can use as reference other articles that used "supplements" with biological properties.
2) format the tables properly, as the first column should not be centered.
3) AER levels in% - in my opinion the authors should use another, more technically acceptable unit - g / kg
4) you used "Lipid Oxidation" - I prefer the term lipid peroxidation.
5) you have results to build a much stronger conclusion. Review her.
These changes do not detract from the merit of your manuscript, but I hope it will help to enhance it.
Author Response

(The authors gave the same response as above.)

Reviewer 4 Report
See attached file.

Author Response
Thank you for your comments and suggestions. We appreciate your responsibility as a reviewer and assistance with our manuscript. You have provided several good suggestions that are very helpful for revising and improving our manuscript, as well as providing significant guidance to our research. We have studied comments carefully and have revised to the best of our ability. Please see the attachment.

This manuscript is a resubmission of an earlier submission. The following is a list of the peer review reports and author responses from that submission.